# Tracking Workplace Violence over 20 Years

**DOI:** 10.3390/ijerph21111438

**Published:** 2024-10-29

**Authors:** Nicola Magnavita, Igor Meraglia, Giacomo Viti, Martina Gasbarri

**Affiliations:** 1Department of Life Sciences and Public Health, Università Cattolica del Sacro Cuore, 00168 Roma, Italy; igor.meraglia01@icatt.it (I.M.); giacomo.viti01@icatt.it (G.V.); 2Local Sanitary Unit Roma4, 00053 Civitavecchia, Italy; martina.gasbarri@aslroma4.it

**Keywords:** health surveillance, aggression, threat, harassment, risk assessment, participatory methods, COVID-19, prevention, longitudinal study, occupational epidemiology, methodology

## Abstract

**Introduction.** Violence against healthcare workers (HCWs) is a widespread, underreported, and inadequately prevented problem. Only a few companies have efficient systems for assessing the extent of the phenomenon. **Methods**. In 2005, the health surveillance service of a public health company introduced a system that monitored violence experienced by HCWs by means of three items from the Violent Incident Form (VIF) integrated with departmental in-depth analyses using the participatory ergonomics group technique. **Results**. In 2005, the annual rate of physical assaults was 8.2%, that of threats was 12.0%, and the harassment rate was 19.6%. Over the past twenty years of observation (2005–2024), the percentage of workers who reported experiencing a physical attack in the previous year at their periodic medical examination has fluctuated between 5.8% and 11.1%, except for the years 2020 and 2021 when, during the COVID-19 pandemic, the rate was 3.9% and 3.2%, respectively. During the same pandemic period, the annual threat rate, which ranged from 9.4% to 20.1%, dropped to 7.7%, while the prevalence of harassment, which was between 13.5 and 19.6, fell to 7.2%. HCWs believe that (i) limiting visitor access, (ii) a better balance of the demand for services, and (iii) a better attitude towards HCWs were the causes of the reduced rate of violence during the pandemic. **Conclusions.** Recording the violence experienced during health surveillance is an economical, reliable, and sustainable risk assessment method.

## 1. Introduction

Workplace violence (WV) against healthcare workers (HCWs) is a long-standing, underreported [1,2,3,4,5] and poorly understood phenomenon. The complex panorama of WV includes physical assaults as well as verbal and psychological violence [6]. These types of violence can manifest themselves as bullying [7], mobbing [8], incivility [9,10,11], discrimination [12], sexual harassment [13], and stalking [14]. The perpetrators are principally patients, especially in cases where they are under the influence of drugs or affected by mental illnesses and cognitive deficits, but they can also be visitors and relatives [15,16,17,18]. Colleagues or superiors are responsible for a non-negligible amount of WV; this so-called “lateral” form of violence is the least spontaneously reported [19,20,21]. Although the consequences of WV may result in obvious physical injuries of varying severity, or even lead to murder [22,23,24,25], those that trigger the emotional involvement of the victim are often the most persistent and harmful [26]. A systematic review found that in HCWs there is a recurring association (resulting mostly from medium-quality research) between physical violence and poor mental health [27] as well as potential connections between psychological violence and poor mental health and sick leave [27]. WV can induce or aggravate anxiety [28,29,30], depression [31,32,33,34], suicidal ideation [35], burnout [36,37,38], post-traumatic stress disorder [39,40], sleep problems [41,42,43], headaches [44,45], musculoskeletal disorders [46,47,48] stress [49,50,51,52,53], poor compassion [54], poor workability [55], absenteeism [56], presenteeism [57], high employee turnover rates [58,59,60], decreased productivity [61], or the worsening of the quality of care [62]. Longitudinal studies have shown that WV causes distress and poor social support, which in turn increase the risk of experiencing WV [63,64].

Although WV is so relevant and widespread, the problem is understudied [65] and lacks a universal definition [6]. International bodies, such as the International Labour Office (ILO) with Convention No. 190 on Violence and Harassment [66] and Recommendation No. 206 [67], the World Health Organization with the Framework Guidelines for Addressing Workplace Violence in the Health Sector [68], and the European Agency for Safety and Health at Work [69], have issued definitions of WV, but the fact remains that, in the perception of workers, this term may have a variable meaning. The same occurs with workplace bullying, where the definition of the phenomenon “is in the eye of the beholder” [70]. A similar attribution uncertainty also exists over mobbing and incivility, which are commonly considered synonyms for bullying, and civility, which is the antonym [71]. In brief, WV is a subjective issue since what is considered “violence” by one employee may be viewed as lower-level antagonism by another [72]. This kind of terminological uncertainty is reflected in the scientific literature and greatly hinders comparisons between different samples, systematic reviews, and meta-analyses and makes it difficult to correctly interpret the problem and propose appropriate solutions. However, from the point of view of occupational medicine, WV is only important if the employee considers it to be so. For this reason, we preferred to focus on the subjective factor by asking the worker if he/she had been the target of violence. Those who did not report cases of violence in their confidential interview with the occupational doctor evidently believed that what happened was not important for their health and their work.

Because it is so commonplace, employees frequently accepted WV as a normal occurrence or an essential component of their jobs [73,74,75,76,77]. Occupational medicine has more than three centuries of history since Bernardino Ramazzini, in his pioneering book “De Morbis Artificum Diatriba” (Diseases of Workers), described the link between the work environment and diseases [78]. For many years, however, researchers did not take WV into consideration. The idea that WV needs to be addressed has arisen only recently in the scholarly literature. Finnish researchers began to include WV among the causes of stress in HCWs only at the end of the 1980s [79], and it took several years before the first data on compensation for assaults and insurance costs appeared in official statistics [80]. Studies derived from the health surveillance of workers in healthcare companies emerged much later, at the beginning of this century, especially in Italian research [81,82,83,84], since the authors of scientific publications preferred to conduct studies on aggregate, national, or supranational data rather than examine the situation of individual working situations [85,86,87,88,89,90,91,92,93,94]. Nowadays, we are seeing the attention of numerous research institutions and organizations, such as the Centers for Disease Control and Prevention (CDC) [95], the American Occupational Safety and Health Organization (OSHA) [96], the European Foundation for the Improvement of Living and Working Conditions (Eurofound) [97], the International Council on Nurses (ICN) [98], and other nursing associations [99]. Recommendations and guidelines provided an overall vision of the phenomenon and the general characteristics potentially manifested in each category of workers but were of limited use in interpreting local situations and preparing suitable prevention measures. Unfortunately, not all companies have occupational epidemiology services capable of effectively assessing these situations. The lack of local data means that risk assessments of WV in healthcare companies are often based on information that refers to totally different conditions. Even worse, risk assessment is often conducted using unvalidated algorithms that provide utterly unreliable results [100]. Since this situation is undoubtedly dangerous for the safety and health of employees and for the repercussions it has on the quality of care, simple and economical methods of monitoring the extent of the phenomenon need to be devised in order to avoid underreporting of the problem and stimulate worker participation. Recommendations on risk assessment and the management of WV call for the development of participative, nondiscriminatory, and systematic strategies [101]. This need is also clear in the recommendations of the International Labour Organization [102].

Italy is currently among the small but growing number of nations that compels employers to evaluate the likelihood of violence, take steps to prevent it, and shield employees from its consequences [103]. In 2007, the Italian Ministry of Health issued a Recommendation [104] that invited all hospital and outpatient facilities to develop prevention programs after analyzing events and work situations. Although many years have passed since the Recommendation was endorsed, it is still not being universally applied in healthcare. All over the world, health organizations are required to conduct violence assessments as part of The Joint Commission’s accreditation guidelines [105].

In this study, we set out to describe observations conducted over the last 20 years in a public health company where the frequency of violent events was measured during health surveillance. The aim of the study is to gain a more detailed knowledge of the phenomenon, ascertain the possible existence of trends, and interpret their causes in order to develop and improve control measures. A secondary objective of this study is to illustrate the method used to monitor violence and discuss its advantages and limitations.

## 2. Materials and Methods

In Italy, workers who are exposed to occupational risks are subjected to mandatory health surveillance in the workplace. This surveillance involves carrying out medical examinations (usually annual) and conducting investigations designed to promptly identify early changes produced by occupational exposure to risk factors. Health surveillance is mandatory but can also include health promotion activities. Traditionally, our university has always combined promotion with prevention by integrating health promotion projects into health surveillance. Although participation in these projects is not obligatory, it is generally very high (over 85%) [106].

In the company where this research was conducted, we started to analyze violent incidents in 1999 using the Violent Incident Form (VIF), a tool specifically created for this purpose by Arnetz [107]. This tool uses an operational definition of violence that encompasses verbal aggression and threats. The questions are “*1. Over the past 12 months have you experienced a physical assault during working hours? (‘physical assault’ means an attack, with or without weapons, that may or may not cause physical harm); 2. Over the past 12 months have you been subject to a threat during working hours? (‘A threat’ refers to the intention of causing physical harm). 3. Over the past 12 months have you been subjected to harassment during working hours? (‘harassment’ means any annoying or unpleasant act (words, attitudes, actions) that creates a hostile work environment)*”. If the worker gives an affirmative answer, he/she is asked to describe the episode by identifying the circumstances, perpetrator, event, and consequences.

Initially, the questionnaire was used within the company to define cases of violence that had been officially reported. However, it soon became clear that the number of reports was very low and limited to the most dramatic incidents, especially if the perpetrator was not a patient. One of the authors of this study (NM), in 2004, also took on the role of Clinical Risk Manager and started a systematic action against violence, defining a policy and starting the training of workers and the collection of WV reports through mailboxes where workers could leave an anonymous report, but responses remained numerically irrelevant. Consequently, to meet prevention requirements and legal indications, in 2005, he began to systematically monitor the violent events that had occurred among workers by asking them the three aforementioned questions during their periodic medical examinations. Over the years, the company implemented a wide range of interventions to combat violence, concerning, on a case-by-case basis, architectural structures, work organization, the replacement of furniture and tools, the activation of alarm and safety measures, the education and training of workers, and other measures that cannot be detailed here.

In the meantime, the health surveillance service continued to contribute to the WV risk assessment through various activities that are not reported in detail in this article. The inspections that the occupational doctor carried out in the various sectors of the company were always integrated by involving Participatory Ergonomics Groups (PEGs) [108], in which the workers described the production cycle, identified its shortcomings, and proposed solutions. During these meetings, the workers discussed the WV problem in their department. Furthermore, participants in GEP meetings were invited to complete the VIF questionnaire online.

In an annual report, the data collected during routine medical examinations and information emerging from the PEGs were brought to the attention of the company management, the company prevention and protection service, and the workers’ safety representatives, thereby contributing to risk assessment and the preparation of WV control measures.

This study refers exclusively to the analysis of three binary questions on physical aggression, threats, and harassment. We calculated year by year the prevalence and used the exact binomial test with the exact Clopper–Pearson 95% Confidence Interval [109,110,111,112] to measure the degree of uncertainty in prevalence. To interpret the trend of violence over the years, the simple linear regression method was used. Statistical tests were performed using IBM SPSS Statistics for Windows, Version 26.0. Armonk, NY, USA: IBM Corp.

The study was conducted in accordance with the Declaration of Helsinki and national laws governing the collection of this kind of data. No ethical opinion is required before collecting medical history data during mandatory medical examinations. At the end of the check-up, after reading the personal risk booklet, the worker declares that what is reported in his/her medical history corresponds to what was declared at every point and that he/she has not withheld information regarding previous or ongoing pathological events. Moreover, he/she also confirms receiving information concerning the meaning of the health surveillance performed and its results, the assessment of suitability for the specific job (the worker is given a copy), and the possibility of appealing against it to the Supervisory Body within the deadline of thirty days. Finally, the worker expresses consent to the management and electronic processing of his/her personal data on the part of the doctor for statistical or scientific purposes, even after the end of the health surveillance period. By signing the personal health document, the worker also agrees to the collective anonymous publication in accordance with the code on the protection of personal data (Law Decree 30/6/2003 n.196) and the principles of the ICOH code of ethics for occupational health operators [113] and on occupational medicine confidentiality principles (Legislative Decree 19/9/1984 n. 626, and Legislative Decree 9/4/2008 n. 81). Furthermore, all health promotion projects were submitted to the competent institutional reviewer boards (Università Cattolica del Sacro Cuore, Fondazione Policlinico Agostino Gemelli, Territorial Ethics Committee) in the years ranging from 2005 to 2024.

## 3. Results

The prevalence rates of workers who declared, during periodic medical examinations, experiencing at least one physical assault, threat, or episode of harassment in the previous year are shown in Table 1.

In 2005, the annual prevalence of physical assaults was 8.2%. Between 2005 and 2016, the average percentage of workers attacked was more than 9%. In other words, in the first decade of observations, more than 1 in 11 workers had been physically assaulted at least once a year. Since 2016, there has been a reduction in the percentage of workers reporting physical attacks (from 7.5% to 6.2%). However, in the three-year period 2016–2018, 1 worker in 15 declared having been physically attacked in the previous year. The pandemic resulted in a significant reduction in the percentage of workers who experienced attacks: the rate fell to 3.9% in 2020 and 3.2% in 2021; in the two-year period, fewer than 1 in 30 workers were assaulted. The frequency of workers reporting physical aggressions showed a decreasing trend, with the pandemic crisis being the lowest point. A linear regression analysis in the period 2005–2021 showed a standardized linear regression coefficient beta = −0.808, with determination coefficient R^2^ = 0.625 (*p* < 0.001) (Figure 1).

Unfortunately, the end of the pandemic period coincided with an increase in the percentage of those who were physically attacked. In 2022, physical violence rose to 5.6%, in 2023 to 6.9%, and in the first semester of 2024, the physical violence rate rose again to that of 2016 (7.5%). From 2021 to 2024, the regression of physical violence rate over time had a standardized regression rate (beta) of 0.950 (Figure 2).

Twelve percent of workers reported having been threatened in 2005. The rate of employees reporting threats remained high in the following years, reaching peaks of over 20%. Between 2005 and 2019, the average rate was over 14%, meaning that one in seven workers had been threatened at least once in the previous working year. The frequency of workers reporting having suffered threats decreased from 2013 until the pandemic period, with a standardized linear regression coefficient beta = −0.850 (*p* < 0.01) (Figure 3). Also, in this case, the pandemic outbreak coincided with a decrease in threats. In 2022, the threat rate halved (7.7%), but the end of the pandemic coincided with a rapid increase in attacks. In 2024, more than 14% reported having been threatened at work. From 2021 to 2024, the rate of workers reporting exposure to threats had a standardized regression beta coefficient over time of 0.857 (Figure 4).

Figure 5 shows the trend in the rates of workers experiencing physical violence and threats over the 20-year period. The COVID-19 pandemic led to the lowest level.

The prevalence of workers who reported experiencing harassment at work (19.6% in 2005) remained consistently high, with an average of 16% in the first 15 years of observation. Only during the pandemic were significantly lower values reported (9.2% and 6.2%), but the frequency quickly rose to values comparable to pre-pandemic ones.

The information collected with the three questions administered during health surveillance was added to the more extensive information collected during the workplace inspections and the PEGs open to all workers in each department.

According to the workers who participated in the departmental PEGs, the reduction in aggression during the pandemic period was mainly due to the filtering of access. All those who wished to enter the healthcare areas had to prove that they were not infectious, and this involved carrying out tests or providing a vaccination certificate. Moreover, the procedure also envisaged verification of the real need to gain access. On the contrary, patients and accompanying visitors had previously been able to indiscriminately enter rooms that were too small to accommodate them and where no one was available to provide information.

Apart from these administrative shortcomings, during the pandemic, many people limited their own access to health areas because they preferred to avoid or postpone non-urgent or non-essential access. Workers in the outpatient departments and those in the surgical areas were the ones who particularly noticed that the lower demand allowed them to work better, with less time pressure and an absence of queues.

In general, the fear of contracting infection led to a reduction in close contact between healthcare workers and patients, even in hospital wards. Workers acknowledged that this reduced the occasions on which workers could be subjected to attacks by patients who were not in full possession of their mental faculties.

Finally, workers reported that in the early stages of the pandemic, their work was regarded with great respect by the population, and this reduced the possibility of uncivilized behavior on the part of patient visitors and relatives. However, as the pandemic continued, the reduction in contact between patients and their relatives increased the latter’s anxiety and gradually people began to fear that the hospital staff were not doing their best. A worsening of opinion towards the health sector was intertwined with fear of vaccinations and medical therapies, thus fueling a state of distrust that provided fertile ground for controversy and criticism, often leading to verbal or physical aggression. The aforementioned crowded conditions in which anxious patients and/or their relatives faced long queues also often led to explosive situations. In some cases, lone workers were left with the impossible task of providing information for the patients or visitors confronting them. A clear example of this was the triage nurse in the emergency room who was frequently attacked for this reason. The workers suggested the company set up a service designed to inform patients of company policies and procedures and the purposes of what was being done.

Overall, HCWs believed that (i) limiting visitor access, (ii) a better balance of demand for services, and (iii) a better attitude towards health workers were the causes of the reduction in the rate of violence during the pandemic. They were concerned that when restrictions ended and visitor access was no longer controlled, relatives and patients would soon recreate the previous situation.

## 4. Discussion

This study reports the results of the prolonged monitoring of physical, verbal, and psychological aggression against HCWs in a public health company using an innovative method based on the collection of information during health surveillance. An analysis of data collected over 20 years demonstrated a gradual but very slow reduction in violence rates, with a more significant reduction in assaults during the COVID-19 pandemic. The trend in violence rates and the characteristics of the method used are discussed below. It is important to underline, however, that this study refers to all the workers of a local health company and consequently reports the average effect between the different departments and services of the company. During the pandemic, in fact, some company services were put to the test and the rate of violence in them did not decrease at all. For example, a study performed in an Italian Emergency Department from January 2017 to August 2021 observed a significant increase in the rate of attacks; trends compared to pre-pandemic months do not seem to indicate a return to normality after the pandemic [114].

### 4.1. Analysis of the Trend in Violence and Measures for Prevention

Before conducting this analysis, we must consider that in the public health company observed, the systematic recording of cases of violence led in 2005 to the adoption of an anti-violence policy and the implementation of risk containment measures. However, for many years, the measures adopted by the company to control the risk of physical violence produced only limited outcomes. A decline in the percentage of workers complaining of physical attacks was gradually observed until a dozen years later, but the number still remained quite high, especially in the psychiatric care and emergency/first aid departments, i.e., the ones at greatest risk. Paradoxically, the rate of physical attacks diminished during the pandemic when the perpetrators were almost exclusively patients who were not in full possession of their mental faculties. We will leave aside the prevention and control interventions of WV implemented by the company over the course of many years, and we will deal with the situation caused by the pandemic because we believe that from this dramatic experience, it is possible to obtain elements for prevention.

Workers witnessed both a reduction and a resurgence in assaults as restrictions put in place during the pandemic were relaxed. We examined in greater detail the causes that led to the paradoxical reduction in WV during the pandemic by recording the opinions of workers during the participatory ergonomics groups organized in each department. Workers attributed the decrease in attacks during the COVID-19 pandemic mainly to the filtering of access to work areas that had limited the presence of visitors and relatives. Another reason indicated for the reduction in WV during the pandemic was the lower demand for services that produced an optimal ratio between staff and patients, thereby improving the quality of care. Moreover, during the very first phase of the pandemic, the general appraisal of healthcare workers on the part of the population further reduced the causes of conflict.

When asked about the organization of their work, healthcare workers were fully convinced that limiting visitor access is a measure that should be universally implemented not only during emergencies. This belief is endorsed in the literature. Field studies demonstrate that unrestricted access to working areas, the lack of security guards and police officers, and the limited intervention on their part are among the causes of WV [115]. To prevent WV in hospital settings, recommended measures include regulating visitor flow, monitoring access, communication initiatives, and training in conflict management and lifesaving [116,117]. Another universal violence prevention measure advocated in the literature is the provision of enough staff to adequately deliver services without time pressure. According to the workers, a better balance of the demand for services and the ability to provide them would be a major deterrent against incivility, because it would eliminate the main cause of grievances that relatives, visitors, and users have towards the healthcare institution. Previous studies indicated that staff shortage could be among the causes of WV, together with third-party misunderstandings of health policy [118,119]. In our study, HCWs also attributed the reduction in WV to a better appraisal of their work in the early stages of the pandemic. They pointed out that the perception (present in the early stages of the pandemic) that HCWs were striving to improve the health of the population, should always be present during the daily activities of the health service. Studies have already indicated that public attitude towards medical staff has an important influence on violence. Many HCWs believe that unfavorable public opinion may be linked to an increase in violence [120]. A study has shown that a percentage of people justify violence, and those who support and excuse violent behavior against medical staff are also more inclined to act aggressively [121].

The prompt identification of potentially aggressive patients could be useful in limiting cases of violence [122]. HCWs are generally thought to need specific training in recognizing and counteracting WV. However, there is almost negligible or poor evidence that interventions focusing on the perpetrator of aggression result in a reduction in WV [123]. In relationships with potentially aggressive patients, training in behavioral skills has been shown to be more efficacious than traditional methods for improving staff performance and competence [124]. To be effective in promoting a safe environment, staff competence must be combined with supportive leadership [125]. Workers’ training is the most frequently proposed measure to prevent WV [126,127,128]. Given the heterogeneity of the educational methods, the effect of this intervention alone is uncertain [129]. A systematic review showed that although education and training can enhance personal knowledge and good attitudes, the latter does not necessarily have an impact on workplace hostility directed toward healthcare staff [130]. A more complex response to WV is needed on the part of health institutions. Warshawski et al. [131] suggested that to reduce WV, policy makers should implement preventive measures such as hiring more medical and nursing staff, providing workshops on handling violence, launching campaigns against violence in healthcare settings, and enforcing suitable punitive measures against attackers. In high-risk departments, such as emergency and first aid, interventions have predominantly involved behavioral training on de-escalation skills or self-defense techniques. Only in a few cases have organizational and environmental interventions been added to these. The effectiveness evidence of these interventions is still sparse [132]. It seems fair to say that only an integrated intervention, which in addition to worker training considers the reorganization of services, the rationalization of spaces and procedures, and the creation of better social relations, can reduce WV in healthcare activities. This opinion seems to be generally accepted because only an integrated strategy on several levels is likely to reduce WV, as was demonstrated by an intervention we conducted over many years in a psychiatric rehabilitation facility [133]. Unfortunately, there are few examples in the literature of the successful application of such measures.

Because the COVID-19 pandemic put healthcare to the test all over the world, even topics such as WV, which were previously of little interest, have received attention. Hundreds of studies have reported the presence of episodes of violence against HCWs. The fact that most of these studies were cross-sectional, usually had no control group, and failed to provide any reference to the previous state of affairs made the results difficult to interpret. Even some systematic reviews and meta-analyses merely reported that WV was prevalent during COVID-19 without indicating whether it was increasing, decreasing, or stable compared to the pre-pandemic era [134,135]. However, other reviews observed an increase in the WV rate between the mid-and late-pandemic phase [136,137], and another synthesis study that included pre-post data confirmed that there was a reduction in violence rates during the pandemic compared to the previous situation [138].

The pandemic was a complex experience for the healthcare workforce and the world population. The rapid evolution of the epidemic and the frantic search for solutions meant that working conditions in hospitals changed several times. Consequently, the relationship between HCWs and other parties and the risk of WV also changed continuously. In the first wave, a lack of knowledge about the new disease and how to treat it, the shortage of personal protective equipment, and the high mortality rate meant that fear for one’s own health and that of one’s loved ones was dominant. Lockdowns limited contact between the population and healthcare personnel. HCWs who had unprotected contact with infectious patients suffered from anxiety and sleep disturbances [139,140,141,142,143,144]. In the COVID-19 hub centers, HCWs suffered mainly from an excessive workload and compassion fatigue [145], while the population considered them heroes [146,147,148,149], or at least expressed gratitude and a favorable opinion. As time passed, the second and third pandemic phases saw HCWs increasingly fatigued due to a high work overload. Depression and burnout in HCWs worsened [150], and the reduction in contact between staff and patients’ relatives worsened relationships and the quality of care [151,152]. Trust in HCWs waned and they were increasingly criticized and isolated [153,154,155], even outside of the hospital. Criticism, discrimination, and WV caused HCWs to experience psychological distress and, ultimately, depressive symptoms [156]. The availability of vaccines brought a gradual easing of restrictive measures, but improvements were slow and the workload for those dealing with COVID-19 remained excessive for a long time. Cases of burnout increased in COVID-19 hub centers, and many quit their jobs [157]. Relationships with visitors and relatives became critical and contacts were increasingly strained [158]. Alongside an increase in social violence during the COVID-19 pandemic, violence against HCWs also became more prevalent [159]. During the fourth phase, HCWs found themselves treating numerous anti-vaxxers [160,161,162] who were sometimes aggressive towards them [163]. All these variations certainly influenced the risk of WV and were not easy to express in a systematic review that did not take prospective evolution into account. However, on the whole, the studies in the literature confirm that the WV trend during the pandemic was similar to the one we observed, and the causes of WV reported by workers in this public health unit are concordant with those reported by workers elsewhere in the world.

### 4.2. Analysis of the Violence Monitoring Method

Monitoring violence during workers’ periodic medical examinations offers many advantages and a few weaknesses. It entails posing only the first three questions of the VIF to workers during their routine medical examination to identify the percentage of those who have experienced assaults. The workers who give affirmative answers may be invited by the doctor to provide further information and possibly to complete the entire questionnaire. The decision to ask only three questions was due to the need to make the duration of the examination compatible with production needs and the obligation to investigate all occupational risks during a single check-up. In this way, there was only a modest time commitment, and no demands were placed on workers who had not been attacked. It is important to remember that by applying the VIF in this way, the outcome represents the annual prevalence of workers who experienced violence, not the frequency of violent episodes. We know from the literature that the extent of WV exposure differs greatly from country to country as well as study location, practice settings, work schedules, and occupation but is generally high, especially in psychiatric and emergency departments and among nurses and physicians [16]. Studying WV prevalence within a health company allows you to follow its evolution over time and also compare different departments, occupational categories, and types of service.

In our opinion, one of the advantages of this type of study is the fact that the conversation with the doctor does not expose the worker to the unwanted consequences that some fear when formal complaints are addressed to the company or the police authority. Fear of victimization [164], a non-supportive culture, and the lack of an efficient and user-friendly reporting system [5] are among the most common causes of non-reporting. Interviewing all workers offers another advantage since it guarantees that none of them will go unheard. If a worker does not report experiencing any violent incident in the previous 12 months, this means that what happened was of no significance.

The subjective nature of the reports we sought could be considered a disadvantage. However, all the reports can be verified by examining the official reports available for the previous year. We deliberately asked about experiences of workplace violence during the preceding 12 months in order to compare self-reporting with official incident reports from the same period. Over the entire observation period, the number of cases reported during medical examinations was always much higher than those recorded as workplace incidents or referred to the authorities. In fact, most of the attacks were reported only to the supervisor and this generally prevented the news from reaching top management, whereas with the system we adopted, the latter was always promptly informed.

The VIF questionnaire focuses exclusively on reporting events in which the worker was personally involved and does not consider the witnessing of episodes of violence, even though it is known that witnessing violence without being a victim can also damage an employee’s health [165,166,167] and may have social consequences that cause violence to appear normal, thus increasing the intention to use verbal or even physical violence [168]. Since WV is a criminal offense and anyone who witnesses an episode of violence is required by law to report it, even if they were not the victim, we decided to exclude this type of assessment. The sole question we asked concerning employees’ experiences with violence at work was whether they had ever been the subject of physical assault, threat, or harassment. Hospital policy requires employees to report any known violent episodes to a supervisor. The policy actually states that any “known incident of violence” should be reported; it does not specify that an employee has to be the actual target of violence before reporting the violent episode.

Recall bias was a possible confounder of this study. Considering that occupational medical examinations are mostly annual, we decided to concentrate on the workers’ experience over the previous 12 months to help them remember whether the episode actually occurred after the previous periodic examination and therefore needed to be reported. This could prevent experiences of violence from being reported more than once. The one-year recall period is the most frequently used in retrospective studies of violence; in this way, it coincides with routine reports on the working environment, which must be made every year by the occupational doctor and the person responsible for the prevention of and protection from risks.

An advantage of the assessment organized in this way is that results are periodically communicated to the employer and other figures who deal with health and safety at work. Employees who verbally report aggression to their supervisors may be meeting their reporting obligations, but top management may not receive these informal reports in time to make policy decisions. Only with the facts at hand can a hospital system or other healthcare organization develop prevention plans.

The systematic collection of information during medical examinations reveals a higher number of violent events than those obtainable from official sources. Official data represent only the tip of the iceberg of violence. The survey conducted in a nearby public health company of the Lazio Region, in the five-year period 2010–2014, collected only 22 reports, corresponding to an annual rate of 0.22 percent [169]. Another large healthcare company operating in a territory partly coinciding with that of our company, after an awareness campaign on the issue of violence and the establishment of dedicated services for victims, in a 34-month period from March 2019 to December 2021, collected only 21 reports, corresponding to an annual rate of 0.25 percent [170]. In a large-sized Italian university hospital in the Lombardy Region, 3.3% of workers had experienced almost one aggression in the 2015/17 three-year period [171]. In two other hospitals in Lombardy, where incident reporting procedures and data collection forms were neither standardized between hospitals nor specific for aggressions, after observing severe underreporting in the period 2016–2020, a new web-based reporting and analysis system was introduced in 2021, thus leading to more reliable results. In this way, the reported prevalence was about 2–3%, with higher levels observed in emergency departments and psychiatric wards [172].

The method we have applied allows us to collect a significant number of incidents, not so serious as to induce the victim to make an official report but still harmful to his/her health. Unlike most published studies, the survey was not conducted only on a professional category or a particularly exposed department and aimed to involve all workers, because both the selection of high-risk groups and the self-selection of participants tend to overestimate the phenomenon. Studies that did not adopt these precautions generally obtained higher prevalence rates than ours. For example, a multicenter cross-sectional online study of Italian HCWs conducted from May 2018 to March 2020 estimated a 10% one-year rate of physical violence [173]. The administration of the VIF to triage nurses in an Italian emergency department resulted in a 96% violence exposure rate [174]. Meta-analysis studies, which collect surveys conducted with very different methods and from many countries, obtained much higher rates of physical violence than those here reported, ranging between 14% and 28% per year [1,16,134,135,175,176,177,178]. An umbrella review of meta-analyses places the annual rate of physical violence to which HCWs are exposed at 20.8% [179]. The threat rate we obtained is also lower than those of the most recent meta-analyses, which estimated rates between 30% and 43% [8,16,175,176]. Similarly, surveys that have investigated verbal violence reported rates much higher than those observed over our long time series; for example, 49.1% of Arabian HCWs [180], and 84.2% of Chinese psychiatric nurses [181] reported one-year verbal violence.

The method used collects only the violent incidents that the worker deems worthy of being reported, often some months after they occurred. The immediate detection of all events can provide higher values. For example, a prospective observational study conducted in 2022 in the emergency department of Merano Hospital, Italy, employing 38 nurses and 11 physicians, recorded 91 violent incidents in an 8-month period, 94% of which were verbal [182].

We must consider that the method used, though minimizing underreporting, also leaves many episodes unreported. The likelihood that the three types of events investigated by the VIF questions will be reported varies: physical assaults are well remembered, especially if the protagonist is not a patient; the same goes for threats, whereas harassment is often not deemed important enough to report. In fact, we sometimes observe that harassment is reported less frequently than threats, which is strange considering that threats are generally associated with other forms of incivility and harassment. Apparently, workers who reported physical violence or the threat of physical harm did not find it necessary to also report harassment.

The method of using the first three questions of the VIF to collect workers’ experiences of WV during health surveillance proved to be economical, sustainable, and accessible. Workers had no difficulty in providing the requested information and the health surveillance activity was in no way burdened by this request since it was always performed during annual health promotion activities.

As in all occupational medicine activities, proof that the questions have been administered to all workers in the same way is guaranteed only by using written questionnaires. For this reason, in the health company studied, the three VIF questions were included in the questionnaires used annually for health promotion activities. The doctor was thus able to analyze the data on violence in the same way as he/she analyzed the other data obtained from health surveillance, all of which, by law, must be presented in an annual report to the employer, the person in charge of prevention and company protection, and workers’ representatives responsible for safety. The very basic information that is the subject of this article was brought annually to the attention of the parties interested in prevention and contributed to determining safety measures. As we have seen, the pandemic had the paradoxical effect of reducing aggression towards medical staff and this could be useful for understanding the organizational measures that might reduce the risk of WV under normal conditions.

The limited information collected via the first three questions of the VIF was integrated with two more essential elements. The first, which consisted of the responses of the workers collectively questioned during the participatory ergonomics groups held in each department at the end of the occupational doctor’s routine medical examination, provided the qualitative data that we presented on the causes of the reduction in attacks during the pandemic. The second entailed inviting workers to fill out the entire VIF questionnaire online in order to describe the most relevant events of which they had been victims. This part of the investigation provided information regarding the circumstances and consequences of individual violent events that were not the subject of this study.

All the data produced by the health surveillance service were made available to workers and their representatives as well as to company management and prevention services to stimulate the adoption of new and better systems of prevention. It is not possible to summarize the practical impact that the data have had on anti-violence policies, because Italian public health companies are subject to continuous and frequent changes in management that obey the election results and the spoil system rather than health needs. Consequently, the development of prevention projects often takes on a stop-and-go trend. However, it is clear that the level of attention to health and safety issues, and violence in particular, has made great progress in the company, and the effort of employment services has probably been the protagonist of this.

### 4.3. Advantages and Weaknesses

One of the strengths of this study is that it was a longitudinal investigation conducted over a 20-year period with an innovative method. However, it also has some weaknesses: performing the study in a single public health company limited the possibility of extending what was observed to other situations, even if the literature indicates that conditions may be similar in other healthcare companies. Another weakness lies in the fact that the respondents were limited only to those who participated in health promotion activities. Nevertheless, it is better to obtain data from a small number of workers rather than have a lack of information resulting from the underreporting of WV. One limitation is certainly the fact that monitoring is based on only three questions; however, the surveillance service and other prevention operators have numerous other means at their disposal to evaluate the causes, consequences, and collateral phenomena of WV.

## 5. Conclusions

The implementation in 2005 of a system for monitoring WV perpetrated against workers in a public health company by means of three VIF questions administered during periodic medical examinations enabled us to analyze the trend in relation to attempts to control this phenomenon. It also revealed that during the COVID-19 pandemic, the percentage of workers who experienced violence dropped sharply and subsequently returned to pre-pandemic levels. When interviewed with the GEP technique on the causes of this paradoxical improvement, workers indicated that the control of access to work areas, providing adequate staffing for service demands, and the then-prevailing favorable attitude of public opinion towards the medical profession were protective factors against violence. Recording WV during health surveillance proved to be an economical, reliable, and sustainable risk assessment method. Collecting the opinions of workers in participatory ergonomics groups conducted in the workplace and obtaining individual responses via the VIF questionnaire regarding the perpetrators, circumstances, and consequences of violent acts provided overall knowledge of this phenomenon that may be of use for developing a policy of well-targeted and effective prevention.

The active collaboration of the health surveillance service in the WV risk assessment that this article has discussed falls within the wider framework of the EU Framework Directive on occupational safety and health, indicating that all employers are required to carry out risk assessment and that it should cover violence and harassment, the provisions in ILO C190 explicitly noting that violence and harassment are workplace hazards that employers need to address through risk assessment and the overall management of health and safety. After too many years during which the problem has been considered marginally, there is, today, a lot of recent activity in this direction in Italy in collective bargaining and through the establishment of a national committee on OSH, which has been exploring new regulations and preventive measures. Health surveillance, with its direct contact with workers, is crucial in these processes.

## Figures and Tables

**Figure 1 ijerph-21-01438-f001:**
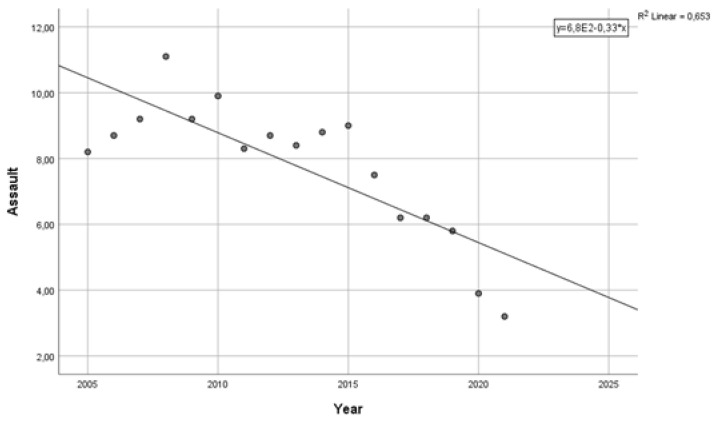
Reduction in the prevalence of workers reporting assaults in the previous year, 2005–2021.

**Figure 2 ijerph-21-01438-f002:**
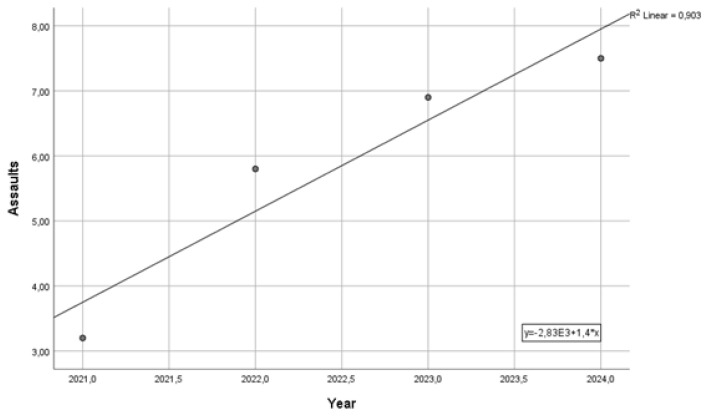
Increase in the share of workers reporting assaults in 2021–2024.

**Figure 3 ijerph-21-01438-f003:**
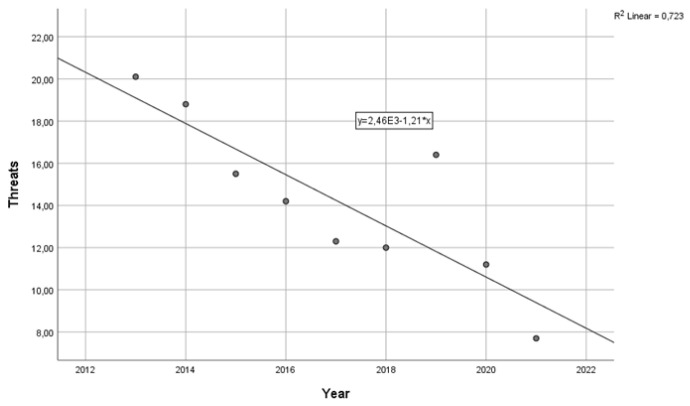
Reduction in the prevalence of workers reporting having received threats in the previous year, 2013–2021.

**Figure 4 ijerph-21-01438-f004:**
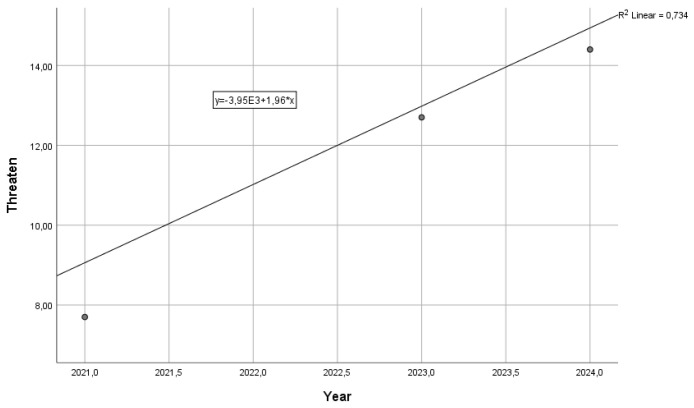
Increase in the share of workers reporting threats in 2021–2024.

**Figure 5 ijerph-21-01438-f005:**
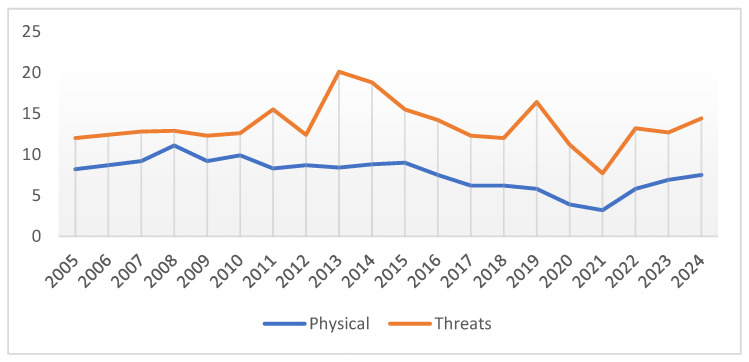
Trends in physical violence and threats over the 20-year period. The arrow points to the COVID-19 pandemic.

**Table 1 ijerph-21-01438-t001:** Prevalence of physical aggressions, threats, and harassment in the 2005–2024 period.

Year	Physical Assaults Prevalence (CI 95%) ^#^	Threats Prevalence (CI 95%) ^#^	Harassment Prevalence (CI 95%) ^#^
2005	8.2 (6.9; 9.7)	12.0 (10.2; 14.1)	19.6 (17.8; 21.5)
2006	8.7 (7.3; 10.3)	12.4 (10.5; 14.6)	19.2 (17.3; 21.2)
2007	9.2 (7.9; 10.7)	12.8 (11.0; 14.9)	18.6 (16.8; 20.5)
2008	11.1 (9.8; 12.6)	12.9 (11.1; 15.0)	18.0 (16.2; 19.9)
2009	9.2 (7.8; 10.8)	12.3 (10.5; 14.4)	19.6 (17.8; 21.5)
2010	9.9 (8.5; 11.5)	12.6 (10.7; 14.8)	13.5 (11.7; 15.5)
2011	8.3 (7.0; 9.8)	15.5 (13.7; 17.6)	15.9 (14.1; 17.8)
2012	8.7 (7.4; 10.2)	12.4 (10.6; 14.5)	14.1 (12.3; 16.0)
2013	8.4 (7.1; 9.9)	20.1 (18.3, 22.2)	18.7 (16.9; 20.6)
2014	8.8 (7.5; 10.3)	18.8 (17.0; 20.9)	17.4 (15.6; 19.3)
2015	9.0 (7.7; 10.5)	15.5 (13.7; 17.6)	14.7 (12.9; 16.6)
2016	7.5 (6.2; 9.0)	14.2 (12.4; 16.3)	13.6 (11.8; 15.5)
2017	6.2 (4.9; 7.7)	12.3 (10.5; 14,4)	15.8 (14.0; 17.7)
2018	6.2 (4.8; 7.8)	12.0 (10.0; 14.3)	13.4 (11.4; 15.5)
2019	5.8 (4.5; 7.3)	16.4 (14.6; 18.5)	13.9 (12.1; 15.8)
2020	3.9 (2.6; 5.4)	11.2 (9.4; 13.3)	9.2 (7.4; 11.1)
2021	3.2 (1.9; 4.6)	7.7 (5.9; 9.8)	7.2 (5.4; 9.1)
2022	5.8 (4.5; 7.3)	13.2 (11.6; 15.3)	12.9 (11.1; 14.8)
2023	6.9 (5.6; 8.4)	12.7 (10.9; 14.8)	12.9 (11.1; 14.8)
2024 *	7.5 (5.7; 9.6)	14.4 (12.0; 17.3)	14.9 (12.4; 17.7)

* Reported in the period between 1 January and 30 June 2024. ^#^ Exact Clopper–Pearson Confidence Interval 95%.

## Data Availability

Data are deposited on Zenodo, https://doi.org/10.5281/zenodo.13342131.

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
