# Peer review of "Tracking Workplace Violence over 20 Years"

_ijerph, 2024, doi:10.3390/ijerph21111438_

Round 1
Reviewer 1 Report
Comments and Suggestions for Authors
- The manuscript addresses a pressing issue of workplace violence (WV) against healthcare workers (HCWs), a topic of global importance, especially in the post-COVID-19 healthcare environment. The longitudinal nature of the study, tracking workplace violence over 20 years, provides unique insights into long-term trends. The connection between violence rates and the COVID-19 pandemic adds a timely and relevant dimension.
However, the focus on a single health company limits generalizability. The paper could be improved in several areas, especially its generalizability, statistical rigour, and depth of data interpretation. While the trends are interesting, the conclusions may not apply broadly to other healthcare settings due to specific organizational or regional policies. The manuscript refers to multiple previous studies by the authors (Magnavita et al.), which makes it less novel. However, it would benefit from including more international or external studies to compare results and provide a broader perspective on how these findings align with global trends.
One significant limitation is the reliance on self-reported data during health surveillance, which is vulnerable to recall bias and underreporting, particularly for less visible forms of violence like verbal harassment. The authors mention this, but stronger measures to reduce bias (such as validation through other sources) could improve credibility. The use of only three questions from the Violent Incident Form (VIF) limits the depth of the data. While the brevity of the questions was chosen for practical reasons, this approach may lead to missing out on more nuanced forms of violence (e.g., emotional or psychological violence). The lack of a control group or comparison with other companies weakens the ability to draw conclusions about the effectiveness of the interventions or the pandemic's unique impact on violence rates.
The manuscript attributes the reduction in violence during COVID-19 mainly to visitor restrictions and reduced demand for healthcare services. However, the broader societal factors, such as stress, fear, and public perception, are underexplored. More in-depth analysis of external factors, beyond the workplace environment, would enhance the interpretation.
The discussion of trends is somewhat superficial. Although trends are reported, little detailed statistical analysis supports the conclusions (e.g., confidence intervals, p-values). The authors could have strengthened their argument by applying more rigorous statistical methods, such as time-series analysis, to validate observed trends.
The practical recommendations are somewhat generic. For example, "limiting visitor access" may not be feasible in many contexts, and "adequate staffing" is a well-known challenge in healthcare. More specific and actionable recommendations would be helpful, particularly regarding how these measures could be implemented in diverse healthcare settings.
The manuscript lacks a discussion on long-term interventions beyond the pandemic. How can healthcare organizations build on these findings to develop sustainable violence prevention policies?
Author Response
Reviewer #1
- The manuscript addresses a pressing issue of workplace violence (WV) against healthcare workers (HCWs), a topic of global importance, especially in the post-COVID-19 healthcare environment. The longitudinal nature of the study, tracking workplace violence over 20 years, provides unique insights into long-term trends. The connection between violence rates and the COVID-19 pandemic adds a timely and relevant dimension.
However, the focus on a single health company limits generalizability. The paper could be improved in several areas, especially its generalizability, statistical rigour, and depth of data interpretation. While the trends are interesting, the conclusions may not apply broadly to other healthcare settings due to specific organizational or regional policies. The manuscript refers to multiple previous studies by the authors (Magnavita et al.), which makes it less novel. However, it would benefit from including more international or external studies to compare results and provide a broader perspective on how these findings align with global trends.
Response: We are very grateful to the reviewer who acknowledged the originality of this study. To the best of our knowledge, there are no companies in the health sector or other areas where violence against workers has been monitored over such a long period of time. Health surveillance of workers is a complex activity that includes many moments in which the doctor comes into contact with workers’ experiences of violence. Medical visits are one occasion, but other occasions include inspections of work environments, meetings of participatory ergonomics groups of workers, health promotion campaigns, surveys on a specific topic and in-depth investigations into reported injuries, which represent a minimal percentage of the total. Naturally, not all of these aspects can be covered in a single article, and this results in the frequency of citations from previous works, which is not a repetition. We had already cited in the previous version over 150 studies, many of which were research and meta-analysis on the topic. We gladly applied the advice and have included three types of citations in this new version: 1) studies conducted on other health companies in the country, so that it is clear that our study can be conveniently compared to other health realities, as is the case for all other field studies; 2) international epidemiological surveys that have obtained prevalence rates of WV on HCWs, to verify the variations that are connected with the measurement method used; 3) experiences of prevention of violence, to give a hint on a topic that is, however, outside the scope of this article.
One significant limitation is the reliance on self-reported data during health surveillance, which is vulnerable to recall bias and underreporting, particularly for less visible forms of violence like verbal harassment. The authors mention this, but stronger measures to reduce bias (such as validation through other sources) could improve credibility. The use of only three questions from the Violent Incident Form (VIF) limits the depth of the data. While the brevity of the questions was chosen for practical reasons, this approach may lead to missing out on more nuanced forms of violence (e.g., emotional or psychological violence). The lack of a control group or comparison with other companies weakens the ability to draw conclusions about the effectiveness of the interventions or the pandemic's unique impact on violence rates.
R.: The aim of this article was to show the trend of violence measured with a constant method throughout the entire observation period (20 years). The literature on workplace violence suffers from extreme variability in measurement methods, ranging from the collection of reports to the use of countless retrospective survey tools. We can say with certainty that employee-submitted reports relate to a very small percentage of actual cases; prevalence rates of reported aggression in health companies are around 1 - 2%, while all retrospective self-report methods obtain prevalences that are 15-20 times higher. As the reviewer correctly observes, any method used (measurement of reports or self-reporting) leaves a quota of events unreported, especially when harassment or incivility is involved. We believe that retrospective survey minimizes the number of unreported incidents, and the annual cadence of the request, during periodic visits, favors the temporal placement of recall: did the event occur before, or after the previous visit? We have dedicated a part of the article to reiterate this concept.
It is important to remember that the three VIF questions were asked before the medical visits, in which the doctor had the opportunity to examine the report in more detail and verify its reliability. In this study we considered only the three questions, leaving aside the fact that workers who had reported events were generally asked to fill out the complete VIF. Furthermore, as we recall in the article, workers were questioned about the violence suffered department by department (participatory ergonomics groups) and through online surveys. In surveys on workers many other aspects have been studied, about their physical and mental health, work-related stress, sleep, anxiety, depression, etc. The results of these studies and other health surveillance interventions, such as individual counseling of victims of violence, are not reproduced in this study.
Since the study design is a 20-year census, there can be no internal control group. Since there are no studies in the literature of such a long duration with which to carry out a control, we were unable to benefit from an external control; however, we have cited the few studies in which there was a monitoring of violence for a few years. Furthermore, we compared the prevalence rates with those in the world literature.
The manuscript attributes the reduction in violence during COVID-19 mainly to visitor restrictions and reduced demand for healthcare services. However, the broader societal factors, such as stress, fear, and public perception, are underexplored. More in-depth analysis of external factors, beyond the workplace environment, would enhance the interpretation.
R.: It was not the authors of the article who proposed this interpretation, but the workers. The article reports on workers' opinions on the decline in assaults during COVID and the recovery after the end of the pandemic. As it is not a study of causes but a census of frequencies, it is limited to reporting observations and comparing them with the literature. We absolutely agree that an in-depth analysis of the causes that led to a reduction in violence against healthcare workers in the first phase of the pandemic requires the analysis of social factors that are not included in this investigation. We have limited ourselves to reporting on the trend in rates of violence, which showed a decline coinciding with the pandemic. Many have gone so far as to describe the evolution, without knowing the previous state of affairs. This study aims to describe the situation many years before the pandemic, the slow reduction of violence rates and the paradoxical effect of the COVID-19 tragedy on WV.
The discussion of trends is somewhat superficial. Although trends are reported, little detailed statistical analysis supports the conclusions (e.g., confidence intervals, p-values). The authors could have strengthened their argument by applying more rigorous statistical methods, such as time-series analysis, to validate observed trends.
R.: We welcomed the suggestion to indicate the confidence limits and to enrich the study with statistical tests. In the first version of the manuscript, we only indicated the values obtained from the observations, which were transmitted with the annual reports for company management, the prevention service and the workers' representatives. In this revised version, we have used the Clopper-Pearson binomial exact test to estimate confidence intervals of the prevalences, linear regression to estimate the trends, and we added some graphs. We believe that this way the article is more readable, and we sincerely thank the reviewer for the advice he/she gave us.
The practical recommendations are somewhat generic. For example, "limiting visitor access" may not be feasible in many contexts, and "adequate staffing" is a well-known challenge in healthcare. More specific and actionable recommendations would be helpful, particularly regarding how these measures could be implemented in diverse healthcare settings.
R.: We thank you for your observation. We should make it clear to readers that these are not our indications, but the statements made by the workers. We must also better explained that the action to prevent violence in the company began in 2004, when the first author was not only the doctor in charge of workers' health but also the clinical risk manager, and that is was divided into numerous interventions that cannot be reported in detail in this article.
The manuscript lacks a discussion on long-term interventions beyond the pandemic. How can healthcare organizations build on these findings to develop sustainable violence prevention policies?
R.: One of the objectives of this article was precisely to encourage the company to continue with greater vigor in the prevention interventions that had led to a reduction in violence, overcoming the phase of resurgence of violence that followed the pandemic. We have underlined this concept in the discussion and in the conclusions.
Reviewer 2 Report
Comments and Suggestions for Authors
- The materials and methods in this study have not been clearly described.
- In some paragraphs, the author includes too many reference sources in sentence clauses, which are deemed unnecessary. The author also uses too many self-citations, totaling 26 references.
Comments on the Quality of English Language
Minor editing of English language required.
Author Response
Reviewer #2
- The materials and methods in this study have not been clearly described.
R.: We sincerely thank the reviewer for having expressed his/her advice in an extremely concise manner (40 words). We have taken care to explain the method used in greater detail.
- In some paragraphs, the author includes too many reference sources in sentence clauses, which are deemed unnecessary. The author also uses too many self-citations, totaling 26 references.
R.: We thank the reviewer for calculating the number of publications. The high number of self-citations is partly due to some unavoidable factors, which are: 1) the fact that the first author has validated in Italian most of the psychodiagnostic questionnaires in use in Italy and that these instruments were used in the article; 2) the fact that the authors of the article have been monitoring the company’s workers for over 25 years and have therefore had the opportunity to evaluate some characteristics of the population studied that deserve to be remembered. 3) The fact that the research group has been active on the topic of violence for more than 30 years and has produced many contributions on the topic. Furthermore, the first author has over 7300 citations on Scopus and has no need to increase his h-index with useless self-citations. Taking all these factors into account, we have reduced the self-citations as much as possible. Self-citations of the 1st author are 10% of the references.
Reviewer 3 Report
Comments and Suggestions for Authors
Please see my suggestions in the attached word document for some minor revisions relating to the context and some questions about the findings.

Author Response
Reviewer #3
Overall remarks This is an extremely useful and well-written article that gives some good results on the prevalence of violence since 2005 to prevent violence against HCWs through an occupational health reporting tool. There are few published results of tools that have been implemented and fewer examples of ones where progress has been made to track changes in preventing violence against health workers. A significant value of this article is that it is longitudinal and tracks changes over 20 years. Dating back to 2005, it is good to have this evidence, notably that violence declined during the COVID-19 pandemic, which included limited visitor access, amongst other reasons.
I have made some line-by-line comments and also highlight some specific contextual issues related to ILO Convention 190, occupational safety and health (OSH) and risk assessment.
R.: We sincerely thank the reviewer, who carefully reviewed the manuscript, providing many useful suggestions; we are particularly pleased that he/she appreciated the effort we have put in many years of work.
Section 1 Line 51 “…lacks a universal definition”. I would strongly suggest that the authors refer to the internationally agreed definition contained in the ILO Violence and Harassment Convention No. 190 (which Italy has ratified). Article 1 defines violence and harassment as: (a) the term “violence and harassment” in the world of work refers to a range of unacceptable behaviours and practices, or threats thereof, whether a single occurrence or repeated, that aim at, result in, or are likely to result in physical, psychological, sexual or economic harm, and includes gender-based violence and harassment; (b) the term “gender-based violence and harassment” means violence and harassment directed at persons because of their sex or gender, or affecting persons of a particular sex or gender disproportionately, and includes sexual harassment.” https://normlex.ilo.org/dyn/normlex/en/f?p=NORMLEXPUB:12100:0::NO::P12100_ILO_CODE:C190
C190 and the accompanying recommendation R206 explicitly state that workplace violence and harassment are occupational safety and health risks and should be prevented through risk assessment. I believe the authors could situate their riskassessment tool in this wider context, which also includes the EU Framework Directive on OSH.
Also of relevance is the WHO’s 2022 Framework Guidelines for Addressing Workplace Violence in the Health Sector, which highlights the importance of preventing and controlling the risks of physical violence and sexual harassment, the need to strengthen trade union roles and contributions to the prevention of violence and harassment and establishing reporting, recording and notification systems. https://www.who.int/publications/i/item/9221134466
R.: We are pleased that the reviewer has brought to our attention an aspect that may be unclear to readers. It is true that international bodies, such as the ILO Violence and Harassment Convention No. 190 (which Italy has ratified) and the WHO's Framework Guidelines for Addressing Workplace Violence in the Health Sector, contain official definitions of WV, but the fact remains that, in the perception of workers, WV may have a variable meaning. We have rewritten the paragraph so that there can be no misunderstandings.
Line 63 - 65: it would be helpful to highlight studies that have shown that many workers consider violence and harassment are “part of the job” and, therefore, do not report it. That does not mean to say that it is not viewed as a problem. See for example global reports ICN et al. report of 2022 and EMO report of 2022, and UK on sexual harassment in nursing (UNISON and Nursing Times 2021), and in emergency services McKay D, Heisler M, Mishori R, Catton H, Kloiber O (Lancet, 2020) It would also be helpful to acknowledge the substantial work carried out on prevention relating to psychosocial risks such as stress and violence and harassment in the health sector (see for example, EU-OSHA) https://osha.europa.eu/sites/default/files/Psychosocial_risk_management_social_care_en .pdf and https://osha.europa.eu/sites/default/files/2021-10/TE4702422ENC_- _Prevention_of_psychosocial_risks_and_stress_at_work_in_practice_EN.pdf
R.: As the reviewer correctly observes, there is a serious problem of underreporting of violence, with only a modest fraction of workers who experience violence reporting it to their supervisors, insurance companies, or the police. We discuss this issue further in the Introduction and the rest of the article, and used the indicated references. The method used, which consists of asking all workers three questions about the violence they have experienced, drastically reduces underreporting. However, if a worker believes that he or she has not experienced violence, we cannot question his or her assessment.
Line 66: The issue has received considerable international attention in part due to the momentum surrounding the adoption of ILO Convention No. 190. It would be worth mentioning this and the fact that many governments have introduced laws and regulations aimed at addressing and preventing violence and harassment.
R.: In describing the problem, we have adopted a historical approach, to make it clear that the problem of violence is not new, but many have discovered it only today. We thank the reviewer for the references he provided us and that we have added to the work
Line 96: Yes Italy has made good progress but suggest revising this to say that Italy is amongst a growing number of countries across the world that has a legal framework on prevention of violence. There are far stronger regulations in Canada, Australia, the Netherlands, Denmark, amongst others. However, a very important development in Italy was the introduction of legal protections under the Italian penal law covering health and social work professionals in 2020.
R.: Following the reviewer's suggestion, we have modified the sentence to indicate that Italy is among the small but growing number of nations that consider it an obligation to prevent violence at work.
Line 86: this may need to be substantiated by referring to ILO’s report on the role of risk assessment in occupational safety and health in preventing violence and harassment, and to highlight the important role of worker/trade unions in this work. See ILO https://www.ilo.org/publications/major-publications/preventing-andaddressing-violence-and-harassment-world-work-through
R.: We have gladly added this indication which confirms what we have already stated.
Section 2 This section is clear and informative about the background to the study.
Section 3 The results are very interesting and valuable. If available, is there some data about gender and also about the prevalence of sexual harassment?
R.: We have addressed these issues in other studies also conducted on workers of this company, but this study refers only to the monitoring of the experience of physical violence, threats and harassment.
Section 4 Compelling discussion and well presented.
Line 355: Suggest changing “undergone violence” to “experiencing violence”.
R.: we have changed.
Line 386: Did the focus on “assault” limit the study?
R.: We explained that the question asked was whether the worker had personally experienced physical assault, threat or harassment, we did not consider the fact of witnessing violence. However, we added in the Limitation section the following statement “One limitation is certainly the fact that monitoring is based on only three questions; however, the surveillance service and other prevention operators have numerous other means at their disposal to evaluate causes, consequences and collateral phenomena of WV.”
Overall, the occupational medicine/health promotion tool can be viewed as one of several risk assessment tools that healthcare providers can use. In the last two sentences, reference is made to how participatory worker consultations could help to elaborate on the causes/who were perpetrators. It might also be relevant to note that participatory and anonymous methods that assure confidentiality are additional ways to take into account the sensitive nature of the problem of violence and harassment and its under-reporting; such participatory methods need to be widely implemented to get a more detailed level of evidence in risk assessment.
R.: We absolutely agree. We are working on another study detailing the methods by which health surveillance can monitor workplace violence.
However, in my view, setting it in the wider framework of occupational safety and health (OSH) is very important since this is the direction that prevention programmes are going, particularly in response to the provisions on OSH in ILO C190 and R206. It would be good to show how the occupational medicine/health promotional model fits into this wider frame. Ideally, and in line with the EU Framework Directive on OSH (all employers are required to carry out risk assessment) and default these should cover violence and harassment); likewise ILO C190 is explicit in noting that violence and harassment are workplace hazards that employers need to address through risk assessment and the overall management of health and safety. There is a lot of recent activity in this direction in Italy in collective bargaining and through the establishment of a national committee on OSH, which has been exploring new regulations relating to employers’ obligations to include violence and harassment in OSH risk assessments. I’m not suggesting a rewrite, but just a couple of sentences to highlight this issue would be useful to showing the importance of occupational medicine/health promotion in the workplace to OSH objectives.
R.: We believe this advice is very valid. We have added these considerations in the final part of the Conclusion.: “The active collaboration of the health surveillance service in the WV risk assessment that this article has discussed falls within the wider framework of the EU Framework Directive on occupational safety and health, indicating that all employers are required to carry out risk assessment and it should cover violence and harassment, and the provisions in ILO C190 explicitly noting that violence and harassment are workplace hazards that employers need to address through risk assessment and the overall management of health and safety. After too many years during which the problem has been considered marginally, there is today a lot of recent activity in this direction in Italy in collective bargaining and through the establishment of a national committee on OSH, which has been exploring new regulations and preventive measures. Health surveillance, with its direct contact with workers, is crucial in these processes.”
For the discussion, it would be interesting to know, if there is any evidence from the health facility, about what the health facility did to respond to the evidence collected – did it instigate new and better systems of prevention, how did the health and safety committee in the facility use the evidence in its wider OSH prevention programmes, and/or what were the perspectives of the workers themselves about the usefulness of the tool and its wider impact?
R.: This is also a challenging topic. All the data produced by the health surveillance service were made available to workers and their representatives, as well as to company management and the prevention services, in order to stimulate the introduction of new and better prevention systems. It is not possible to summarize the practical impact of the data on anti-violence policies because Italian public health companies are subject to continuous and frequent changes in management, which is more oriented towards electoral results and the spoil system rather than health needs. Consequently, the development of prevention projects often takes on a stop-and-go trend. However, it is clear that the level of attention to health and safety issues, and violence in particular, has made great progress in the company, with the efforts of employment services probably playing the main role.
My final question for the authors is whether trade unions were involved in the project. The unions in Italy have been very active in the health sector, including negotiating sectoral CBAs in health that address violence and harassment. Perhaps this is the subject for a different article, but in case there was union involvement, it would be good to mention this.
R.: The first author of this article has always been active in the medical unions and in 1997-2001 had the honor of representing the Italian unions in Luxembourg in the ad hoc Group "Occupational Exposure Level" of the European Commission, Directorate-General V (Labour, Social Relations and Social Affairs, Public Health and Occupational Safety) in Luxembourg, in the development of Occupational Exposure Limits (OEL) for chemical pollutants in the workplace. For this reason, the collaboration of the unions has always been sought and obtained, even if it has never taken on a formalization.
Round 2
Reviewer 2 Report
Comments and Suggestions for Authors
no more revisions are needed